# Changes of faecal bacterial communities and microbial fibrolytic activity in horses aged from 6 to 30 years old

Marylou Baraille[1,2]*, Marjorie Buttet[2], Pauline Grimm[2], Vladimir Milojevic[3], Samy Julliand[2], Véronique Julliand[1]

1 Institut Agro Dijon, Université de Bourgogne Franche–Comté, PAM UMR A 02.102, Dijon, France, 2 Lab To Field, Dijon, France, 3 Sandgrueb-Stiftung, Zürich, Switzerland

* marylou.baraille@lab-to-field.com

**Data Availability Statement:** All relevant data are within the paper and its Supporting Information files. All raw sequence read files are available from

## Abstract

Horse owners and veterinarians report that from the age of 15, their horses can lose body condition and be more susceptible to diseases. Large intestinal microbiome changes may be involved. Indeed, microbiota is crucial for maintaining the condition and health of herbivores by converting fibres into nutrients. This study aimed to compare the faecal microbiome in horses aged from 6 to 30 years old (yo), living in the same environment and consuming the same diet, in order to assess whether the parameters changed linearly with age and whether there was a pivotal age category. Fifty horses were selected from the same environment and distributed across four age categories: 6–10 (n = 12), 11–15 (n = 11), 16–20 (n = 13), and 21–30 (n = 14) yo. All horses had no digestive problems, had teeth suitable for consuming their feed, and were up to date with their vaccination and deworming programmes. After three weeks of constant diet (ad libitum hay and 860 g of concentrate per day), one faecal sample per horse was collected on the same day. The bacterial communities' richness and intra-sample diversity were negatively correlated with age. There was a new distribution of non-beneficial and beneficial taxa, particularly in the 21–30 yo category. Although the faecal concentration of short-chain fatty acids remained stable, the acetate proportion was negatively correlated with age while it was the opposite for the proportions of butyrate, valerate, and iso-valerate. Additionally, the faecal pH was negatively correlated with age. Differences were more pronounced when comparing the 6–10 yo and 21–30 yo categories. The values of the parameters studied became more dispersed from the 16–20 yo category onwards, which appeared as a transitional moment, as it did not differ significantly from the younger and older categories for most of these parameters. Our data suggest that the microbiome changes with age. By highlighting the pivotal age of 16–20, this gives the opportunity to intervene before individuals reach extremes that could lead to pathological conditions.

the NCBI database (accession number: PRJNA1090926).

**Funding:** The study was funded by the Sandgrueb Foundation, which focuses on animal welfare. The funder Sandgrueb-Stiftung (animal protection foundation) contributed to the study design, data analysis and preparation of the manuscript.

**Competing interests:** The authors have declared that no competing interests exist.

## Introduction

Within the last decade, the proportion of horses aged over 20 years increased from 7% to 12% in France [1,2], from 8% to 17% in Switzerland [3,4], and from 8% to 11% in the USA [5]. In the same time, horse owners and veterinarians report that from the age of 15, their horses can lose body condition and be more susceptible to infectious or chronic diseases [6–8]. Due to a lack of related data, the recommendations for preventive healthcare measures in elderly horses are often extrapolated from those used in adult horses [6]. To provide relevant recommendations, particularly nutritional ones, it is essential to understand certain physiological mechanisms linked to the ageing process in horses [9].

The large intestinal (LI) microbiome plays a key role in digestion, contributing to maintaining the condition and health of the host. Indeed, only microbes are able to degrade fibre which mainly compose the herbivore diet [10]. This degradation enables the release of microbial metabolites (i.e. short chain fatty acids (SCFAs) and lactate). The latter are sources of energy for the horse, and also support various mechanisms that promote good immunity [11,12].

Research into changes of the human LI microbiome that occur with age has been ongoing for the last two decades and is mainly approached by targeting the faecal bacterial communities. The faecal bacterial communities of elderly people appear less diverse and rearranged compared to those of healthy adult people [13–17]. Most studies agree that the abundance of pathobionts increases (i.e. *Desulfovibrio*, *Enterobacteriaceae*, *Eggerthella*) whereas some commensal taxa carrying essential functions disappear (i.e. *Roseburia*, *Faecalibacterium*, *Bifidobacterium*) [18–27]. These changes linked with ageing have also been implicated with the development of frailty [28–30] and diseases such as Alzheimer [31] and Parkinson [32,33]. From about 20 studies included in different reviews [13,14,16,17], only 5 compared microbial activity parameters between elderly and younger people [34–38]. Among these, solely 3 report differences between age categories. In the faeces of elderly people, the percentage of dry matter (DM), the concentrations of tryptophan, indole, iso-valerate, iso-butyrate, iso-capronate, L-lactate and pentadecylic acid are lower while the concentrations of propionate, palmitic and arachidic acids are higher in comparison with younger people [34,36,38].

If the composition of the LI bacterial communities erodes over the lifespan in horses as it does in humans, the LI microbial activity, particularly the fibrolytic activity, may be altered, leading to clinical issues reported by owners [6]. However, in horses, changes in the faecal bacterial communities with age are still little documented. Currently, 2 out of 5 available scientific publications related to this topic mention a decrease in bacterial communities diversity [39,40], whereas 3 highlight a change in bacterial communities structure [40–42] with age. Among these changes, there appears to be a higher abundance of non-beneficial taxa (i.e. *Proteobacteria*, Amplicon Sequence Variant (ASV) assigned to the *Eggerthellaceae* family) and a lower abundance of potentially beneficial taxa (i.e. genus *Fibrobacter*, ASV assigned to the *Ruminococcaceae* family) in elderly horses compared to younger ones. [40,41]. Finally, among these 5 studies, 2 have also investigated microbial activity parameters [41,42]. One of these reports a higher faecal pH in the elderly compared to adult horses [41], while the second observes no difference between the same age categories [42]. In both studies, no difference is observed in faecal SCFAs concentrations between the investigated age categories. These scattered initial results make it difficult to draw conclusions on the involvement of the LI microbiome in clinical signs that appear during the ageing process in horses.

As the age at which a horse is deemed old remains unclear, we decided to follow a cohort of horses aged from 6 to 30 years old (yo), living in the same environment and consuming the same diet. Using a linear approach, our first aim was to assess the change of the faecal bacterial communities with age, while evaluating parameters of microbial fibrolytic activity. Moreover,

in human research, classifying people according to their age is frequently applied, especially since the category of centenarians is studied to identify markers of extreme longevity. Therefore, based on age categorisation, our second aim was to evaluate whether there was a pivotal age in horses from which some of the parameters studied differed markedly.

## Materials and methods

The protocol was approved by the local animal experimentation ethics committee of the University of Burgundy (Comité d'éthique de l'expérimentation animale—Grand campus Dijon N ˚105) and met all requirements for ethical care and treatment of animals.

### Experimental design, animals' characteristics, and management

This study was conducted in July 2020 at a horse sanctuary located in France. Fifty horses aged from 6 to 30 yo, either mares or geldings, of different breeds were included in the study. These animals did not suffer from any digestive issue for at least 3 months prior to the study and were up to date with their deworming and vaccination programmes. Potential metabolic disorders were not tested. At least 5 weeks prior to sampling, horses were submitted to a dental examination and received dental treatment, if needed. Following this, the veterinary dentist scored the wear levels of incisors and molars on a 6-point scale ranging from "no wear" to "very significant wear", and evaluated if each horse could have gripping or chewing problems. Horses' weight was measured and body condition score (BCS) according to the Henneke scale [43] was determined by a single competent examiner to ensure standardised and objective assessment. Characteristics of the horse population are available in the supplementary data (S1 Table).

All horses had been housed in the same sanctuary for at least one year prior to the study. For the study, they were housed in groups, in dry lot paddocks. All horses were fed hay *ad libitum* and received an individual ration of 860 g of concentrate (Landmüsli–Typ Senior, Futtermühle Tock GmbH, Germany) once a day, for 3 weeks to habituate and stabilise the LI microbiome. The biochemical composition of hay and concentrate (Equine Complete; DairyOne, Ithaca, USA) is available in the supplementary data (S2 Table). During this period, they had free access to water and mineral salt block and their general condition was followed daily by the sanctuary's caretakers.

### Sample collection

After 3 weeks of constant diet, freshly voided faecal samples were obtained from each horse once on the same day. For each sample, the central portion was aliquoted in sterile microtubes and immediately frozen at -20˚C and then -80˚C to perform bacterial 16S ribosomal RNA gene (rRNA) sequencing analysis.

For each faecal sample, 10 g of faeces were weighed and dried at 70˚C for 72 hours to determine the faecal DM. Faecal particles size was determined by sieving 150 g of faeces with water through 3 decreasing size sieves (2; 0.5 and 0.15 mm). The proportion of particles for each category was determined and related to DM. Faecal samples were filtered through a 100 μm nylon screen. The faecal pH was measured in the filtrate with a Cyberscan 500 pH-meter (Eutech Instruments, Strasbourg, France). After that, the filtrate was sampled in microtubes with or without a preservative solution (4.25% $H_3PO_4$ and 1.0% $HgCl_2$) and frozen at -20˚C to determine SCFAs and lactate concentrations respectively.

### Bacterial 16S rRNA gene sequencing analysis

Faecal total DNA was extracted as described by Yu and Morrison [44]. Briefly, to lyse the cells, 0.25 g of faecal sample was bead-beaten with a mixture of sodium dodecyl sulfate (SDS), NaCl,

and EDTA. Impurities and SDS were removed by precipitating them with ammonium acetate. Nucleic acids were then recovered by precipitating them with isopropanol. Genomic DNA was purified by sequentially digesting with RNase and Proteinase K, followed by the use of QIAamp columns. After spectrophotometric assessment of the quantity and purity of the DNA obtained (Eppendorf spectrophotometer, Hamburg, Germany), the V3-V4 hypervariable region of the 16S rRNA gene was amplified and sequenced as described by Grimm et al. [45]. Briefly, 2 consecutive polymerase chain reactions (PCR) were performed (PCR 1 for V3-V4 region amplification and PCR 2 to ligate Illumina adapters and index for sample identification). The PCR mix contained DNA, buffer, dNTPs, Taq polymerase and primers (PCR 1: F343 and R784; PCR 2: forward primer `AATGATACGGCGACCACCGAGATCTACACTCTTTCC CTACACGAC` and reverse primer `CAAGCAGAAGACGGCATACGAGAT`-Index-`GTGACTGGAGT TCAGACGTGT`). The PCR programme was as follows: 1 minute at 94˚C, 30 (PCR 1) or 12 (PCR 2) x [94˚C for 1 minute, 65˚C for 1 minute and 72˚C for 1 minute] and 10 minutes at 72˚C. The correct V3-V4 region amplification after PCR 1 was verified by electrophoresis on a 2% agarose gel. The PCR products obtained were sequenced using an Illumina MiSeq run of 250-paired ends, according to the manufacturer's instructions (Illumina Inc., San Diego, CA, United States) at Genotoul Bioinformatics Platform (Toulouse, France). 16S rDNA sequences were submitted to the NCBI Sequence Read Archive and can be found with the following accession number: PRJNA1090926.

FROGS (Find Rapidly OTU with Galaxy Solution) metabarcoding pipeline on the Galaxy server was used to perform bioinformatics analysis [46]. The first step was to assemble the raw data (R1 and R2 reads) and sort them to remove aberrant sequences, i.e. sequences without primers or out of range (<380 or >490 base pairs). Clusters were formed from the remaining sequences using the SWARM aggregation technique (distance = 1). Chimeric sequences were eliminated, and a filter was applied to retain only clusters present in at least 2 samples and with an abundance greater than or equal to 0.005% to obtain the ASV abundance table. Each ASV was aligned to the silva138.1 16S database using BLAST and was affiliated to the highest taxonomic possible rank. Finally, only affiliations with a percentage of identity and coverage greater than 90% and 99% respectively were retained for data analysis.

Firstly, the relative abundance of each bacterial phylum and genus was calculated by relating their abundance to the number of total sequences in the sample concerned. This allowed comparisons of abundance between all the samples. Secondly, the sequences of all samples were rarefied to the smallest number of sequences obtained. Based on this new dataset, the richness (i.e. number of observed ASV and Chao 1 index), intra-sample diversity (i.e. Inverse Simpson and Shannon indexes) and inter-sample diversity (i.e. Bray-Curtis distance) were calculated.

## Determination of short chain fatty acids and lactate concentrations

SCFAs concentrations were determined as described by Jouany [47]. Briefly, filtered faecal samples were injected, under nitrogen, onto a 30 m x 0.25 mm diameter x 0.25 μm capillary column (Elite-FFAP column, PerkinElmer, Courtaboeuf, France) of gas-liquid chromatography coupled to a flame ionisation detector (Clarus, PerkinElmer, Courtaboeuf, France). The internal standard added to all filtered faecal samples was 4-methyl valeric acid (277827-25G, Sigma-aldrich, USA). A standard solution was used to determine the concentration of acetate (C2), propionate (C3), butyrate (C4), iso-butyrate (iC4), valerate (C5), and iso-valerate (iC5) in each filtered faecal sample. The addition of each SCFA gave the total SCFAs concentration. Each SCFA was expressed as a percentage of the total SCFAs.

D- and L-lactate concentrations were measured using an enzymatic colorimetric assay kit (Megazyme International Ltd, Wicklow, Ireland), according to the manufacturer's instructions

and the modifications described by Grimm et al [48]. Optical density was measured at 340 nm (MRX Revelation Microplate Reader, Dynatech Laboratories, Guyancourt, France).

## Statistical analysis

The homoscedasticity of the quantitative variables as a function of age was tested using a Fligner-Killeen test and linear regressions were plotted with a 95% confidence interval using R software to see if there was a linear change of some parameters with age. Pearson correlations between all the parameters studied and age were obtained using the PROC CORR procedure in SAS software.

Furthermore, to determine whether the studied parameters evolved differently from a pivotal age, four categories were compared: 6 to 10 yo (n = 12), 11 to 15 yo (n = 11), 16 to 20 yo (n = 13), 21 to 30 yo (n = 14). Gripping and chewing problems and wear levels of incisors and molars were compared between categories using a Fisher exact test on R software. For quantitative variables, the means of each category were compared using the PROC MIXED procedure (LSMEANS/PDIFF option with Tukey-Kramer adjustment) in the SAS software. Finally, principal coordinate analysis (PCoA) based on Bray-Curtis distances was performed on R software using the Phyloseq package to graphically evaluate the clustering of individual faecal bacterial communities by category. The impact of categories on the structure of the bacterial communities was evaluated by a permutational multivariate ANOVA (PERMANOVA). Galaxy Toulouse platform was used to perform a Linear discriminant analysis effect size (LEfSe) [49], which combined the Kruskal-Wallis sum-rank test to identify taxa with significant differences in abundance between categories (using all-against-all comparisons) and a linear discriminant analysis (LDA) to estimate the effect size of each differentially abundant taxon (with a threshold set at 3 log LDA scores).

For all statistical tests, changes and differences were considered significant at $P \leq 0.05$.

The software versions used are R 4.3.0 (The R Foundation for Statistical Computing) and SAS 9.3 (Statistical Analysis System Institute Inc, Cary, North Carolina).

## Results

All horses remained in good condition and did not develop any health problems during the 3 weeks prior to faecal samples collection.

## Dental check-up of horses and faecal particle sizes

There were no differences in prehension and mastication between age categories. However, the wear level of incisors and molars differed between the categories. Incisors were more worn in horses from 16 to 20 yo and 21 to 30 yo than in horses from 6 to 10 yo and 11 to 15 yo. Horses from 16 to 20 yo and 21 to 30 yo had more worn molars than horses from 6 to 10 yo (Table 1, S1 Fig).

The proportions of different faecal particle sizes were not correlated with age, except for those between 2 mm and 0.5 mm which were negatively correlated (r = -0.29, $P$ = 0.038). No differences in proportions were identified between categories for other particle sizes (S3 Table).

## Faecal bacterial communities

Of the 50 faecal samples, 5 were excluded from the bioinformatic analysis due to a number of sequences less than 2000 after filtering steps. In the remaining 45 samples, 719,225 sequences from the 16S rRNA V3-V4 region were recovered, with an average of 15,983 ± 8,467 sequences per sample. Following various filtering processes, 2,558 ASVs were identified. Rarefaction

**Table 1. Results of Fisher's exact test used to compare gripping and chewing abilities and wear level of incisors and molars between age categories.**

| | *P* |
|---|---|
| Gripping | 0.238 |
| Chewing | 0.255 |
| Wear level of incisors | **<0.001** |
| Wear level of molars | **0.040** |

*P* values in bold indicate that there were significant differences between age categories.

curves (S2 Fig) showed that, at ASV level, some samples had not plateaued, suggesting that a full sampling of these environments had not been achieved.

The richness (number of ASVs and Chao 1 index) and the intra-sample diversity (Inverse Simpson and Shannon indexes) were negatively correlated with age (Fig 1). Among the categories, horses from 21 to 30 yo had a lower number of ASVs compared to those from 6 to 10 yo and 11 to 15 yo (Fig 2A). No difference between categories was observed for the Chao1 index (Fig 2B). The Inverse Simpson index was lower in horses from 21 to 30 yo in comparison to those from 11 to 15 yo (Fig 2C). The Shannon index was lower in horses from 21 to 30 yo in comparison to those from 6 to 10 yo and 11 to 15 yo (Fig 2D).

PERMANOVA analysis highlighted a difference in inter-sample diversity between categories (*P*<0.001) and the PCoA shows the extent of these variations (Fig 3).

The 2,558 ASVs were assigned to 8 phyla, 14 classes, 28 orders, 48 families and 97 genera. Some ASVs have been multi-affiliated. The mean relative abundance of unknown ASVs was 0.2 ± 0.3% at the order level, 1.2 ± 0.9% at the family level and 30.6 ± 2.7% at the genus level. After analysing the phyla, we targeted the genera as they best approximate the functional potential of the bacteria. Correlation with age and the difference between categories were only tested on taxa with an average relative abundance greater than 0.1%.

Of the 8 phyla, 7 had relative abundances greater than 0.1%. Of these, 2 were negatively and 2 positively correlated with age (Table 2). Among the latter, only *Bacteroidota* (*P* = 0.035) was different between categories, with a lower relative abundance in horses from 21 to 30 yo than in those from 11 to 15 yo (S4 Table).

Of 97 genera, 50 had relative abundances greater than 0.1%. Of these, 11 were negatively and 7 positively correlated with age (Table 3). Among the latter, some differed between categories (S4 Table). *Agathobacter* (*P* = 0.025) was less abundant in horses from 21 to 30 yo than in those from 11 to 15 yo. *Butyvibrio* (*P* = 0.013) and *Prevotella* (*P* = 0.015) were less abundant in horses from 21 to 30 yo than in those from 6 to 10 yo. *Desulfovibrio* (*P* = 0.018), *Lachnospiraceae ND3007 group* (*P* = 0.003), *Prevotellaceae Ga6A1 group* (*P* = 0.003) and *Roseburia* (*P* = 0.026) were less abundant in horses from 16 to 20 yo and 21 to 30 yo than in those from 6 to 10 yo. On the other hand, *Christensenellaceae R-7 group* (*P* = 0.004) was more abundant in horses from 21 to 30 yo than in those from 6 to 10 yo and 11 to 15 yo. *Rikenellaceae RC9 gut group* (*P* = 0.009) was more abundant in horses from 21 to 30 yo than in those from 6 to 10 yo. *Lachnospiraceae UCG 009* (*P* = 0.020) was more abundant in horses from 16 to 20 yo than in those from 6 to 10 yo. Finally, *2* genera that were not correlated with age, nevertheless showed differences between categories: *Anaerovorax* (*P* = 0.004) was more abundant in horses from 21 to 30 yo than in those from 6 to 10 yo and 16 to 20 yo and *Oribacterium* (*P* = 0.045) was less abundant in horses from 21 to 30 yo than in those from 11 to 15 yo.

In addition, we used LEfSe analysis to identify bacterial taxa that were strongly associated with each category compared with the others. The differences between categories are illustrated in a circular cladogram (Fig 4) and the LDA scores are grouped in the supplementary

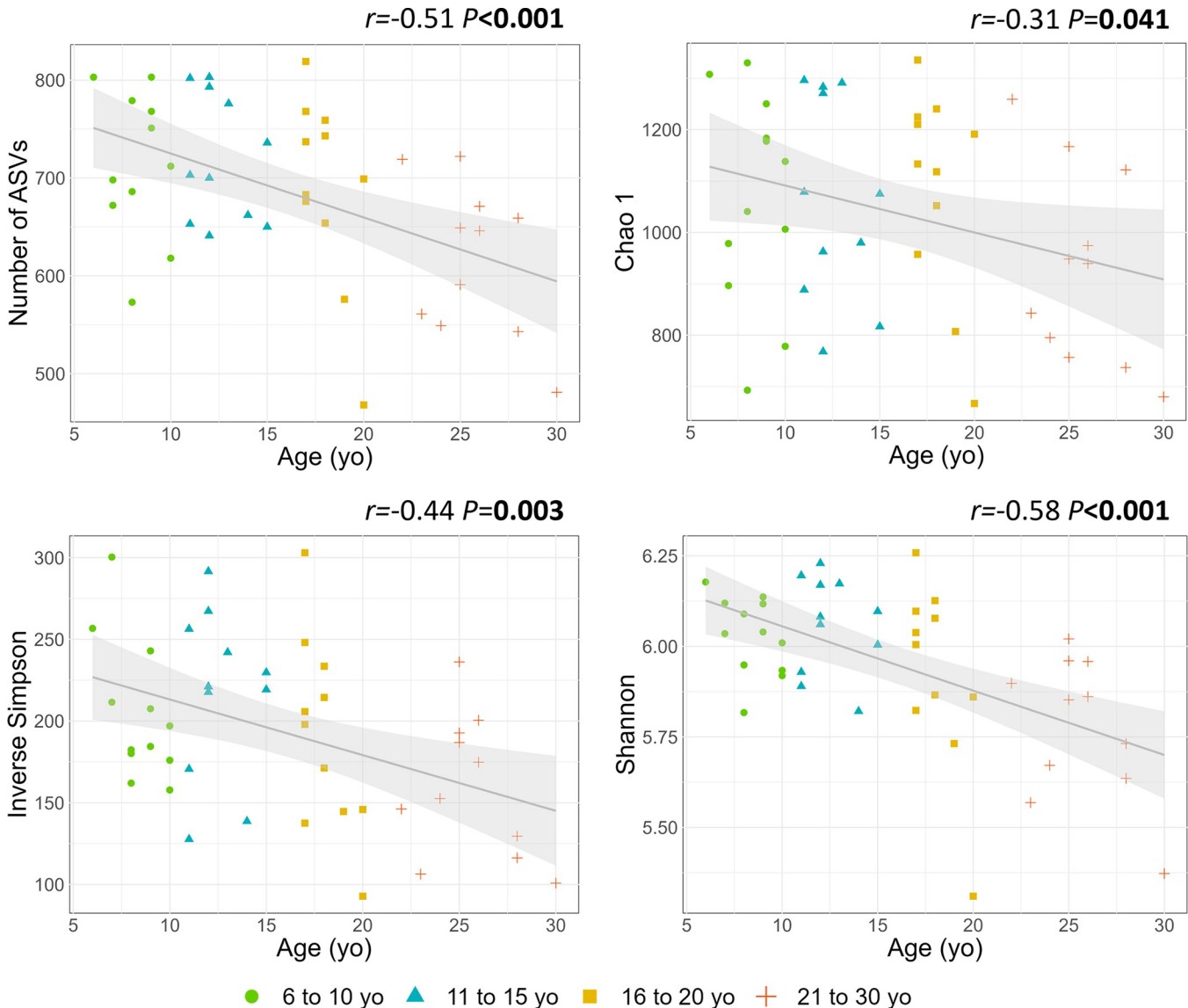

**Fig 1.** Linear regressions illustrating the correlations between faecal bacterial richness (A: Number of ASVs; B: Chao 1 index) and intra-sample diversity (C: Inverse Simpson index, D: Shannon index) and age. Shaded areas represent 95% confidence intervals.

data (S5 Table). In horses from 6 to 10 yo, the class *Lachnospirales* and the families *Bacteroidales RF16 group* and *Lachnospiraceae* were overrepresented. In horses from 11 to 15 yo, the faecal microbiome was enriched in *Bacteroidales* class, *Bacteroidales BS11 gut group* and *p-251-o5* families and *Prevotellaceae Ga6A1 group*, *Lachnospiraceae UCG-003*, *Mailhella* and *Oribacterium* genera. In horses from 16 to 20 yo, only the *Muribaculaceae* family differentiated this class from the others. Finally, the 21 to 30 yo category was distinguished from the others by an enrichment of the *Proteobacteria* phylum, the *Gammaproteobacteria* order, the *Ruminococcaceae* family and the *Enterorhabdus* genus.

## Faecal microbial fibrolytic activity

Faecal DM was positively (Fig 5A), and pH negatively (Fig 5B) correlated with age. There was no correlation between the total SCFAs (Fig 5C) and lactate (Fig 5D) concentrations and age.

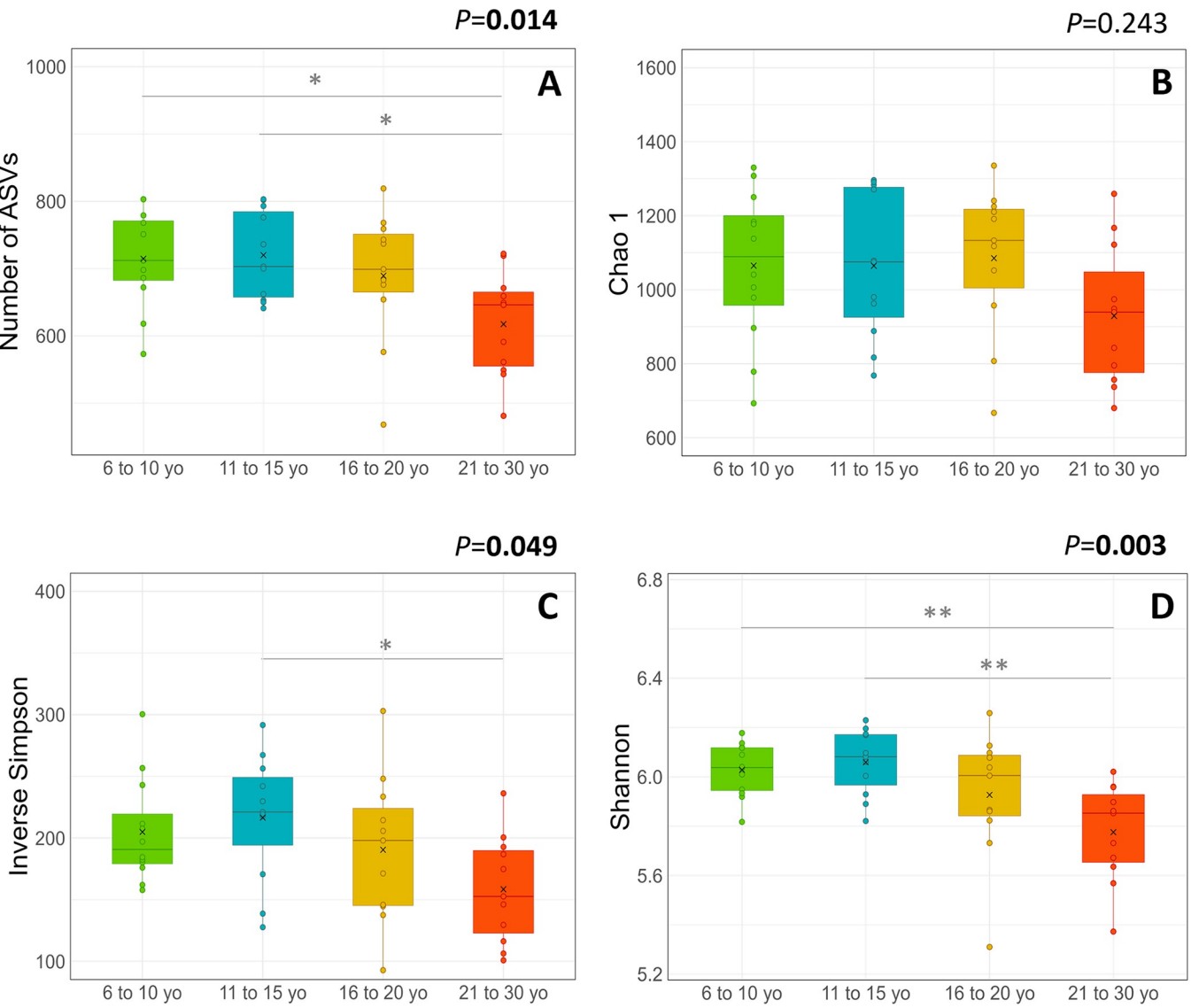

**Fig 2.** Faecal bacterial richness (A: Number of ASV; B: Chao 1 index) and intra-simple diversity (C: Inverse Simpson index, D: Shannon index) in horses according to the age category. The mean (cross), median (solid line) and interquartile ranges are indicated in each boxplot. The $P$ values reported correspond to the comparison of each parameter between age categories. Asterisks indicate significant differences between two age categories (*: $P < 0.05$; **: $P < 0.01$).

When looking at the detail of the fermentation products, the faecal proportion of C2 was negatively and the faecal proportions of C4, iC5 and C5 were positively correlated with age (Table 4). There was no correlation between the C3, iC4, D- and L-lactate proportions and age (Table 4).

In terms of categories, faecal DM was greater (Fig 6A) and pH was lower (Fig 6B) for horses from 21 to 30 yo in comparison to those from 6 to 10 yo. The total SCFAs and lactate concentrations did not differ between categories (Fig 6C and 6D).

When looking at the detail of the fermentation products, the proportion of C4 was higher in the faeces of horses from 21 to 30 yo compared to all other categories (Table 4). The proportion of C5 was higher in the faecal sample of horses from 21 to 30 yo compared to those from 6 to 10 yo and 11 to 15 yo (Table 4). Additionally, the faecal proportion of iC5 was higher for

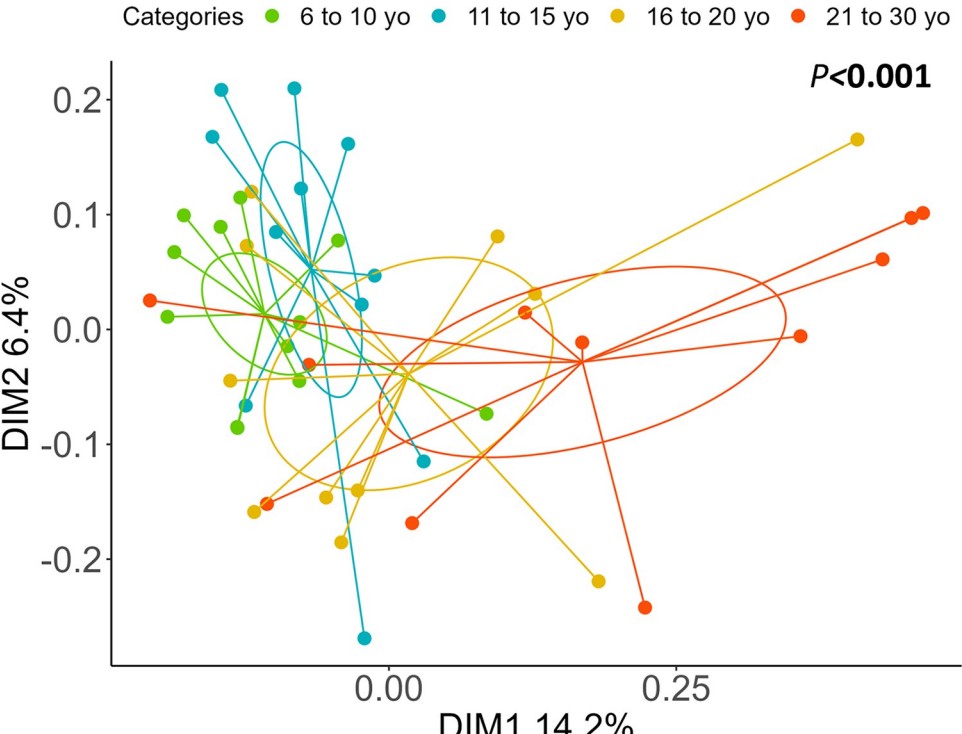

**Fig 3. Principal coordinate analysis (PCoA) at the ASV level showing inter-sample diversity (Bray-Curtis distance) in horses according to age category.** Ellipses represent 95% confidence intervals.

horses from 21 to 30 yo than those from 6 to 10 yo (Table 4). There were no differences between the categories for C2, C3, IC4, D- and L-lactate proportions (Table 4).

## Discussion

This study compared the faecal microbiome (i.e. faecal bacterial communities and microbial fibrolytic activity) of horses aged from 6 to 30 yo, to determine whether the parameters change linearly with age and whether there was a pivotal age category beyond which differences between age categories become more pronounced.

In humans, the country [50] and residence place [28] in which elderly people live, are described as a co-factor contributing to the observed differences in faecal bacterial communities. In horses, the location explains 6.4% of the variation in faecal bacterial communities [40], but not specifically in elderly individuals. In addition, reviews summarise and highlight the major effect of the diet on the large intestinal microbiota [51,52]. Thus, to limit the impact of

**Table 2. Correlations (Pearson correlation coefficients, r) between age and faecal bacterial phyla relative abundances.**

|  | r | P |  | r | P |
|---|---|---|---|---|---|
| *Actinobacteriota* | 0.39 | **0.008** | *Firmicutes* | 0.39 | **0.009** |
| *Bacteroidota* | -0.42 | **0.004** | *Proteobacteria* | 0.16 | 0.289 |
| *Desulfobacterota* | -0.35 | **0.018** | *Spirochaetota* | -0.02 | 0.921 |
| *Fibrobacterota* | 0.03 | 0.823 |  |  |  |

*P* values in bold indicate a significant correlation with age.

**Table 3. Correlations (Pearson correlation coefficients, r) between age and faecal bacterial genera relative abundances.**

| | r | P | | r | P |
|---|---|---|---|---|---|
| *Acetitomaculum* | -0.18 | 0.224 | *Lachnospiraceae UCG-004* | 0.12 | 0.438 |
| *Agathobacter* | -0.32 | **0.034** | *Lachnospiraceae UCG-006* | -0.13 | 0.398 |
| *Alloprevotella* | 0.19 | 0.218 | *Lachnospiraceae UCG-008* | -0.31 | **0.036** |
| *Anaerovibrio* | 0.04 | 0.794 | *Lachnospiraceae UCG-009* | 0.37 | **0.013** |
| *Anaerovorax* | 0.27 | 0.072 | *Lachnospiraceae XPB1014 group* | 0.43 | **0.003** |
| *Blautia* | -0.44 | **0.002** | *Ligilactobacillus* | -0.02 | 0.877 |
| *Butyrivibrio* | -0.51 | **<0.001** | *Marvinbryantia* | -0.13 | 0.401 |
| *Candidatus Soleaferrea* | -0.17 | 0.262 | *Monoglobus* | 0.16 | 0.307 |
| *Christensenellaceae R-7 group* | 0.48 | **0.001** | *NK4A214 group* | 0.34 | **0.024** |
| *Clostridium sensu stricto 1* | 0.30 | **0.049** | *Oribacterium* | -0.23 | 0.134 |
| *Colidextribacter* | -0.09 | 0.544 | *Papillibacter* | 0.19 | 0.208 |
| *Coprococcus* | -0.29 | 0.051 | *Prevotella* | -0.45 | **0.002** |
| *Defluviitaleaceae UCG-011* | -0.19 | 0.214 | *Prevotellaceae Ga6A1 group* | -0.50 | **0.001** |
| *Desulfovibrio* | -0.32 | **0.034** | *Prevotellaceae UCG-001* | -0.05 | 0.765 |
| *dgA-11 gut group* | -0.10 | 0.513 | *Prevotellaceae UCG-003* | -0.33 | **0.029** |
| *[Eubacterium] hallii group* | 0.18 | 0.239 | *Prevotellaceae UCG-004* | 0.25 | 0.099 |
| *[Eubacterium] ruminantium group* | -0.39 | **0.009** | *Pseudobutyrivibrio* | -0.12 | 0.418 |
| *Family XIII AD3011 group* | 0.29 | 0.055 | *Rikenellaceae RC9 gut group* | 0.45 | **0.002** |
| *Fibrobacter* | 0.03 | 0.824 | *Roseburia* | -0.32 | **0.031** |
| *hoa5-07d05 gut group* | -0.10 | 0.528 | *Ruminococcus* | -0.23 | 0.133 |
| *Lachnoclostridium* | 0.06 | 0.683 | *Saccharofermentans* | 0.02 | 0.914 |
| *Lachnospiraceae AC2044 group* | 0.31 | **0.038** | *Streptococcus* | 0.29 | 0.057 |
| *Lachnospiraceae FCS020 group* | 0.21 | 0.172 | *Treponema* | -0.06 | 0.719 |
| *Lachnospiraceae ND3007 group* | -0.50 | **0.001** | *UCG-002* | 0.21 | 0.164 |
| *Lachnospiraceae NK4A136 group* | -0.24 | 0.111 | *UCG-005* | -0.14 | 0.372 |

*P* values in bold indicate a significant correlation with age.

the location and diet and highlight only the effect of age on the faecal bacterial communities, we selected horses kept at one location for at least 1 year and fed the same diet. Our results demonstrated that the faecal bacterial richness and intra-sample diversity were negatively correlated with age, confirming recent observations [40]. Additionally, intra-sample diversity indexes were lower in the 21 to 30 yo category compared to the younger categories. These results are in accordance with data from a previous study comparing a category of horses aged between 19 and 29 with a category of horses aged between 9 and 12 [39]. In humans, it has been suggested that a loss of diversity favours the expansion of the abundance of non-beneficial taxa [17], and both are strongly associated with an unhealthy ageing. It has been linked with increased frailty [29,30,53,54] and cognitive decline [55], as well as Parkinson's [33] and Alzheimer's [31] diseases. In horses, there is no grid to define frailty like those used in humans which are based on parameters such as appearance of illnesses (physical and mental) and painful conditions, taking of medication, reduction in grip strength and speed of movement, history of hospitalization, loss of body condition and autonomy [29,54]. Thus, when examining the effect of age in horses, it is still not possible to attribute the observed microbiome changes to frailty. Screening for metabolic diseases such as insulin dysregulation or Cushing's syndrome could provide interesting information. Insulin dysregulation has a greater impact than age of ponies on the faecal bacterial communities [41]. In contrast, in ponies negative for Cushing's syndrome, when two groups aged 21.5 ± 2.9 years and aged 9.8 ± 3.2 years are

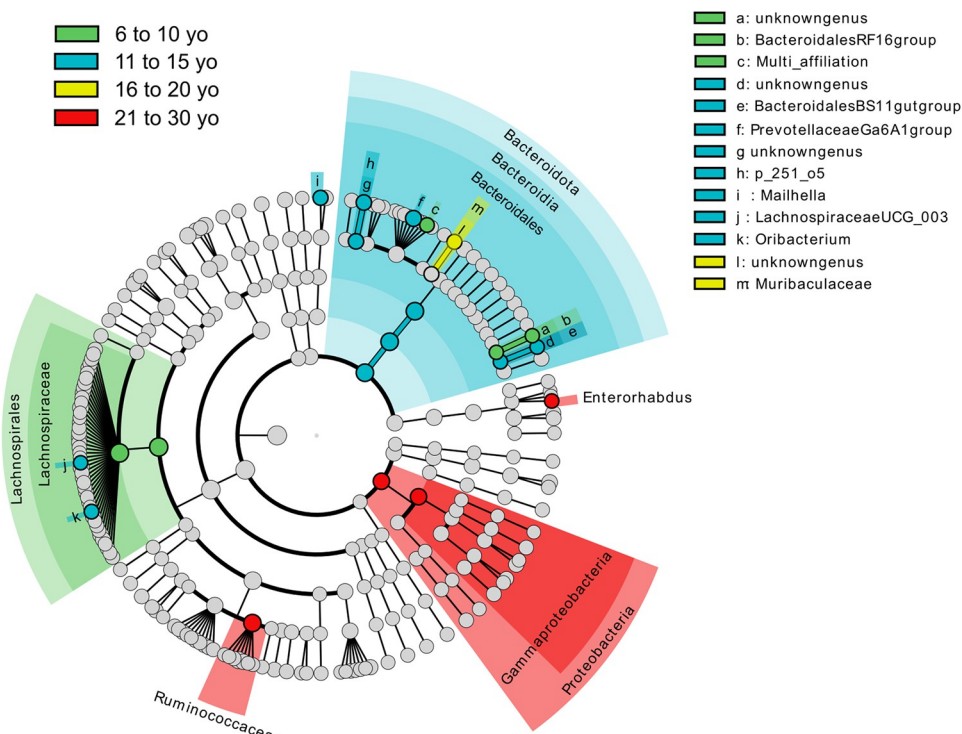

**Fig 4. Cladogram of the LEfSe analysis showing overrepresented faecal bacterial communities as a function of age category.** From the centre outwards, the concentric circles represent the taxonomic level of phylum, class, order, family, and genus respectively.

compared, there is no difference in intra-sample diversity and faecal microbial activity [42]. There is no existing data on the impact of Cushing's syndrome on the bacterial communities in horse faeces. In humans, such metabolic disorder greatly reduces the intra-sample diversity and modifies the structure. Simultaneously, this is accompanied by an increase in the abundance of pathobionts (i.e. *Proteobacteria* including *Escherichia-Shigella*) and parallel reduction of the abundance of beneficial taxa (i.e. *Blautia* and *Agathobacter*), and a lower faecal propionate concentration [56]. The numerous correlations between the parameters studied in our study and age suggested that the changes observed in the microbiome were at least partly linked to ageing, although it cannot be guaranteed that they were linked to healthy ageing. The human healthy ageing microbiome is still in the early stages of being described [17,57]. In healthy elderly people, the decrease in the abundance of important taxa (i.e. *Prevotella*, *Faecalibacterium*, *Coprococcus*) and the increase in the abundance of pathobionts (i.e. *Eggerthella*, *Streptococcus*, *Enterobacteriaceae*) is compensated by the increase in the abundance of other beneficial taxa (i.e. *Akkermansia*, *Christensenellaceae*, *Butyrivibrio*). These taxa disappear when the elderly individual shifts from a healthy state to a state of physiological decline [57]. In our study, a rearrangement also appeared to be taking place, particularly in the 21 to 30 yo category. Simultaneously with the expansion of taxa known to carry pathobiont species (i.e. *Clostridium sensu stricto 1* and, *Proteobacteria* including *Gammaproteobacteria*), there was a rearrangement in the abundance of certain beneficial taxa: *Agathobacter*, *Blautia*, *Butyrivibrio*, *Lachnospiraceae ND3007*, *Prevotella*, *Prevotellaceae Ga6A11 group*, and *Roseburia* make way for *Christensenellaceae R-7 group*, *Rikenellaceae RC9 gut group* and *Ruminococcaceae*. This rearrangement could be responsible for changes in essential functions carried out by the

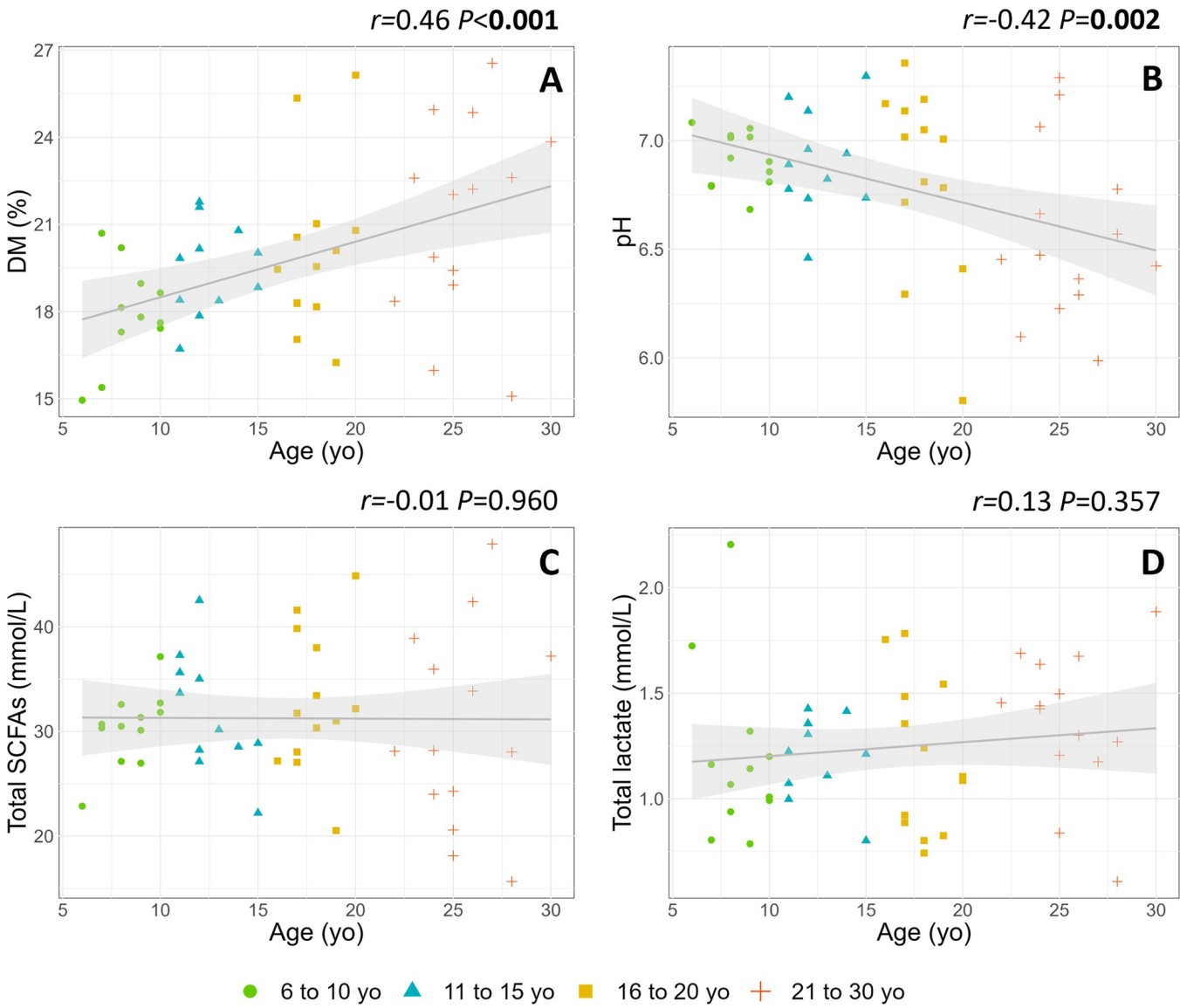

**Fig 5.** Linear regressions illustrating the correlations between faecal microbial activity parameters (A: DM; B: pH; C: Total SCFAs; D: Total lactate) and age. Shaded areas represent 95% confidence intervals.

microbiome, in particular the degradation of dietary fibre, which provides energy and supports host health. To assess this, we measured fibrolytic activity in faeces.

In horses, poor dentition, i.e. an incorrect occlusal angle in the premolar area, reduces apparent fibre digestibility [58] and dental correction improves their digestibility [59]. To avoid bias in the present study, the horses' dentition was checked and treated. Inevitable differences in the wear of incisors and molars between age categories were highlighted, but the ability to grip and chew was similar between age categories as judged by the veterinary dentist. Wear issues did not appear to affect fibre digestibility between age categories. Faecal concentrations of total SCFAs and lactate remained stable with age, confirming previous observations [41,42]. However, acidification of the faecal environment was suggested by the negative correlation between pH and age. Acidification in culture-based experiments is known to have a major impact on some beneficial fibrolytic species and fibre degradation [60]. In our study, we

**Table 4. Correlations (Pearson correlation coefficients, r) between age and faecal SCFAs and lactate proportions, and LSMean of each proportion according to the age category.**

| | Correlation with age | | LSMeans per category | | | | | |
|---|---|---|---|---|---|---|---|---|
| | r | P | 6 to 10 yo | 11 to 15 yo | 16 to 20 yo | 21 to 30 yo | SEM | P |
| **Proportion of total SCFAs (%)** | | | | | | | | |
| C2 | -0.35 | **0.014** | 75.25 | 74.25 | 73.15 | 72.63 | 0.94 | 0.161 |
| C3 | -0.04 | 0.794 | 15.89 | 16.33 | 17.40 | 15.47 | 0.73 | 0.221 |
| C4 | 0.51 | **<0.001** | 5.27[a] | 5.39[a] | 5.36[a] | 6.80[b] | 0.33 | **0.002** |
| iC4 | 0.12 | 0.400 | 1.59 | 1.71 | 1.67 | 1.80 | 0.14 | 0.696 |
| iC5 | 0.40 | **0.004** | 1.51[a] | 1.70[a,b] | 1.70[a,b] | 2.24[b] | 0.18 | **0.014** |
| C5 | 0.60 | **<0.001** | 0.50[a] | 0.62[a] | 0.72[a,b] | 1.06[b] | 0.11 | **0.001** |
| **Proportion of total lactate (%)** | | | | | | | | |
| L-Lactate | -0.12 | 0.412 | 59.36 | 60.19 | 59.75 | 59.30 | 1.82 | 0.982 |
| D-Lactate | 0.12 | 0.412 | 40.64 | 39.81 | 40.25 | 40.70 | 1.82 | 0.982 |

*P* values in bold indicate that there was a significant correlation with age or a significant difference in LSMean between age categories. For each row, LSmeans with different superscripts differ (*P* < 0.05).

C2: Acetate, C3: Propionate, C4: Butyrate, iC4: Iso-butyrate, iC5: Iso-valerate, C5: Valerate. LSMeans: Least square means. SEM: Standard error of the mean.

found that the proportion of faecal acetate was negatively correlated with age while it was the opposite for the proportions of butyrate, valerate and iso-valerate. This suggested that, despite the acidification of the faecal environment, the function of fibre degradation did not decline with age, but rather reorganised, probably due to the rearrangement of bacterial communities that were no longer able to degrade the same type of fibre. It would have been valuable to measure which type of fibre was the most degraded as a function of age, based on enumeration of fibrolytic microorganisms or measurement of fibre digestibility, in order to draw conclusions on this point.

In the 21 to 30 yo category, the pH was lower than in the younger categories. In this category, there was a higher proportion of butyrate, valerate and iso-valerate without a significant decrease in other acids. Butyrate is particularly important as it is essential for maintaining the integrity of the intestinal mucosa, by nourishing the colonocytes that make it up [61]. In our study, this raises the question of a higher production of butyrate and/or an accumulation of this SCFA due to a lower absorption by the intestinal mucosa. At this point, our results are contradictory and do not allow us to conclude. Indeed, on the one hand, we measured a negative correlation with age of some butyrate-producing taxa such as *Roseburia* [62] and *Agathobacter* [63]. Such a decline is consistent with the human literature [64]. On the other hand, the faecal dry matter was positively correlated with age, which could reflect more water absorption in the large intestine, and consequently more SCFAs absorption, since water absorption is always accompanied by SCFAs absorption in the large intestine of horses [65]. However, faecal dry matter modification could also be related to a longer transit time [65] or to a lower water consumption. To confirm the changes in SCFAs absorption with age, it might be informative to measure blood SCFAs. Measurement of the transit time and the daily water consumption could also be valuable data.

In humans, the composition of the microbiota changes gradually throughout life, with no specific chronological threshold at which changes occur suddenly [66]. In horses, our data suggest the same pattern as we observed that many parameters of the faecal bacterial communities and microbial activity were correlated with age. However, even if the change appeared to be gradual, the inter-individual variability was increasingly observed, with visually greater variations from the 16 to 20 yo for several parameters (i.e. PCoA, pH, dry matter, total SCFAs).

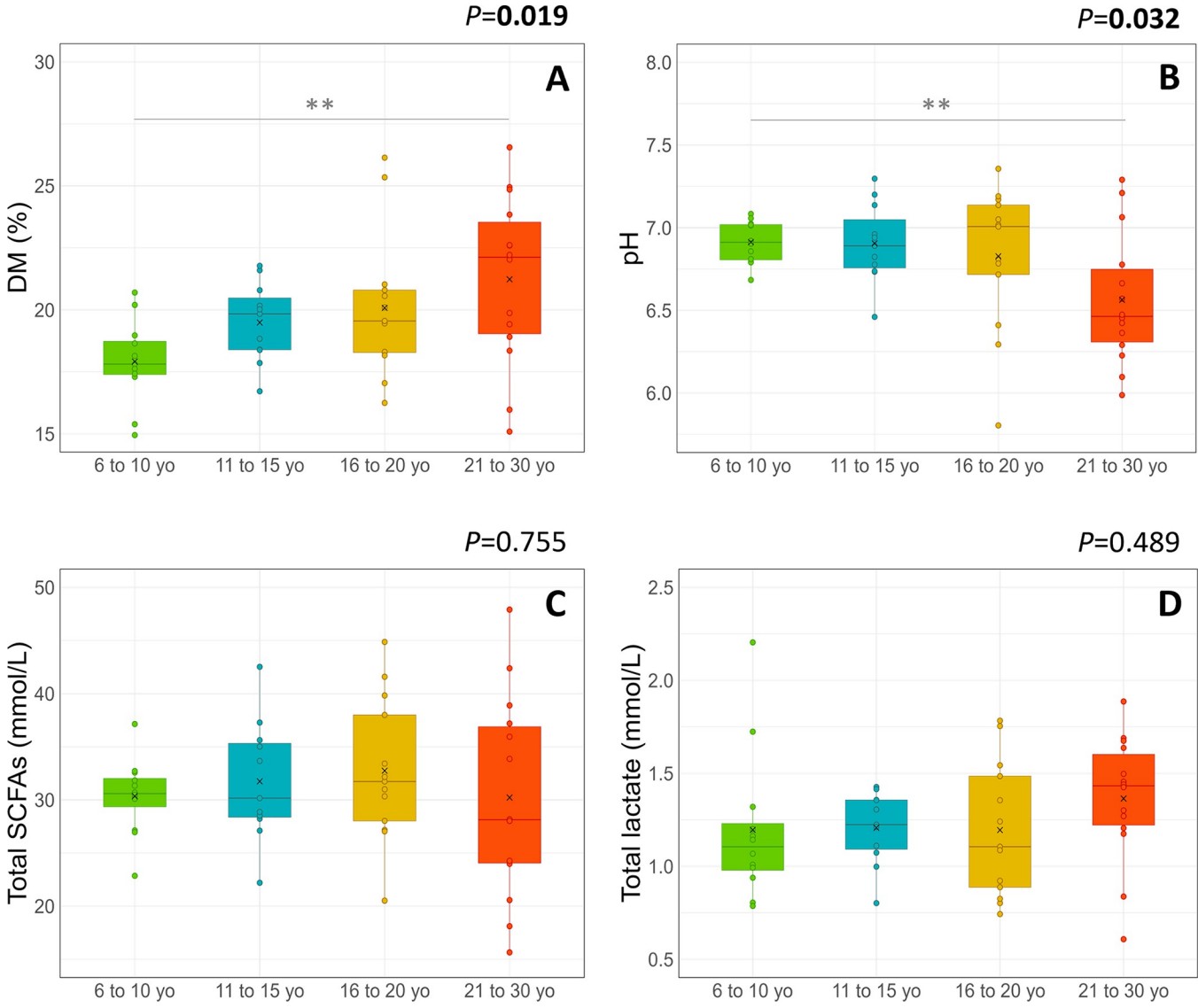

**Fig 6.** Faecal microbial activity parameters (A: DM; B: pH; C: Total SCFAs; D: Total lactate) in horses according to the age category. The mean (cross), median (solid line) and interquartile ranges are indicated in each boxplot. The *P* values reported correspond to the comparison of each parameter between age categories. Asterisks indicate significant differences between two categories (**: *P* < 0.01).

This observation is consistent with findings in humans, where a greater inter-individual variation in faecal bacterial composition in elderly people is observed compared with younger ones [67]. This suggests that changes do not follow the same slope for all individuals, which can explain why some individuals reach a very advanced chronological age in good health, while others lose health earlier. In addition, the 16–20 yo category did not differ significantly from the younger ones (6 to 10yo and 11 to 15 yo) and the older one (21 to 30 yo) regarding most of the microbiome parameters we monitored. It is therefore possible that this 16–20 yo category represented the point at which individuals made the transition towards a pathological state or extreme longevity for the end of their lives. This could explain why some horse owners report more clinical problems after the age of 15 [6,7].

## Conclusion

In horses aged between 6 and 30 years, which have lived in the same environment for a long time and have teeth good enough to consume their feed, we showed that the bacterial communities lose richness and diversity and reorganise themselves with age. Fibrolytic activity in faeces did not appear to be reduced, but also reorganised with age. However, the lack of data on the state of health (i.e. frailty, metabolic disorders) of horses did not allow us to claim that these changes were solely age-related. Defining the parameters of the microbiome of elderly horses that have aged in good health would make it possible to discern the change in the faecal microbiome that predisposes to a pathological condition. Finding out whether there was a pivotal age at which it was relevant to study the horse microbiome was a major challenge. To our knowledge, there is no published data on this topic. From 16 to 20 yo, the values of the microbiome parameters became increasingly scattered, suggesting that from this point on, some individuals reached extremes. This new understanding may allow for intervention before this pivotal age, for example through an adapted diet or by administering biotics (including prebiotics, probiotics, and postbiotics), to influence the microbiome change towards healthy ageing and avoid pathological conditions reported by horse owners.

## Supporting information

**S1 Fig. Gripping and chewing ability and wear level of incisors and molars by age category.** The *P* values reported correspond to the comparison of each parameter between age categories. Asterisks indicate significant differences between two age categories (*: $P < 0.05$; **: $P < 0.01$, ***: $P < 0.001$).
(TIF)

**S2 Fig. Rarefaction curves at the amplicon sequence variant level showing the depth of sequencing of the faecal bacterial communities (obtained after the last filtering step before data normalisation).**
(TIF)

**S1 Table. Individual characteristics of the horses involved in the study.**
(XLSX)

**S2 Table. Biochemical composition of hay and concentrate.**
(XLSX)

**S3 Table. Correlations (Pearson correlation coefficients, r) between age and proportions of particle sizes in faeces, and LSMean of each proportion according to the age category.** *P* values in bold indicate that there was a significant correlation with age or a significant difference in LSMeans between age categories. LSMeans: Least square means. SEM: Standard error of the mean.
(XLSX)

**S4 Table. LSMean values of each phylum and genus relative abundance that were significantly correlated with age (Table 3) and/or differed between age categories.** *P* values in bold indicate that there was a significant difference in LSMeans between age categories. For each row, LSmeans with different superscripts differ ($P < 0.05$). LSMeans: Least square means. SEM: Standard error of the mean.
(XLSX)

**S5 Table. Linear discriminant analysis scores of the bacterial taxa enriched in the faecal microbiome of different age category.** *P* values in bold indicate the dominance of the taxa in

the age category concerned. LDA: Linear discriminant analysis.
(XLSX)

## Acknowledgments

We thank the technical staff for the animal care and laboratory analyses (Gut Aiderbichl team in Trévol, Lab To Field team in Créancey, PMB team from the PAM UMR of the Institut Agro Dijon in Dijon).

## Author Contributions

**Conceptualization:** Marylou Baraille, Marjorie Buttet, Pauline Grimm, Vladimir Milojevic, Samy Julliand, Véronique Julliand.

**Data curation:** Marylou Baraille, Marjorie Buttet.

**Formal analysis:** Marylou Baraille, Marjorie Buttet, Pauline Grimm, Samy Julliand.

**Funding acquisition:** Vladimir Milojevic, Samy Julliand.

**Investigation:** Marjorie Buttet.

**Methodology:** Marjorie Buttet, Pauline Grimm, Vladimir Milojevic, Samy Julliand, Véronique Julliand.

**Project administration:** Marjorie Buttet, Vladimir Milojevic, Samy Julliand.

**Resources:** Marjorie Buttet, Vladimir Milojevic, Samy Julliand.

**Supervision:** Marjorie Buttet, Vladimir Milojevic, Samy Julliand, Véronique Julliand.

**Visualization:** Marylou Baraille, Marjorie Buttet.

**Writing – original draft:** Marylou Baraille, Marjorie Buttet.

**Writing – review & editing:** Marjorie Buttet, Pauline Grimm, Vladimir Milojevic, Samy Julliand, Véronique Julliand.

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
