## [Decision Letter · Decision Letter 0]

29 Feb 2024

PONE-D-23-43107Changes of faecal bacterial communities and microbial fibrolytic activity during the ageing process in horsesPLOS ONE

Dear Dr. Baraille,

Thank you for submitting your manuscript to PLOS ONE. After careful consideration, we feel that it has merit but does not fully meet PLOS ONE’s publication criteria as it currently stands. Therefore, we invite you to submit a revised version of the manuscript that addresses the points raised during the review process.

We look forward to receiving your revised manuscript.

Kind regards,

Franck Carbonero, PhD

Academic Editor

PLOS ONE

Journal Requirements:

"The study was funded by the Sandgrueb Foundation, which focuses on animal welfare."

**Additional Editor Comments:**

The reviewers agree that the manuscript is generally sound, and that mainly minor changes are needed to improve it.

Reviewers' comments:

Reviewer's Responses to Questions

**Comments to the Author**

1. Is the manuscript technically sound, and do the data support the conclusions?

Reviewer #1: Yes

Reviewer #2: Yes

2. Has the statistical analysis been performed appropriately and rigorously? 

Reviewer #1: Yes

Reviewer #2: Yes

3. Have the authors made all data underlying the findings in their manuscript fully available?

Reviewer #1: Yes

Reviewer #2: No

4. Is the manuscript presented in an intelligible fashion and written in standard English?

Reviewer #1: Yes

Reviewer #2: Yes

5. Review Comments to the Author

Reviewer #1: Given the great attention paid to the well-being of the elderly horse, it is of fundamental importance to know what changes occur with advancing age and in what age range they take place, in order to guarantee adequate cares.

This is an interesting study that investigates changes in the microbiome as the horse ages.

I have provided specific comments in the work, as follows:

Introduction:

L 77 “evolution” in “change”.

L 77-78: I would change the sentence “while including parameters of microbial fibrolytic activity” in “and evaluated parameters of microbial fibrolytic activity”.

Materials and Methods:

L 88-97: Do horses, especially older ones, suffer from any metabolic disorder? Specify in the materials and methods that no tests have been carried out to investigate the possible presence of metabolic disorders in elderly subjects.

L 99-100: Have you carried out analyzes of the hay and feed concentrate used? Is it possible to provide a table with the chemical composition of feeds and the nutritional values of the overall daily diet?

L 105: Were faecal samples collected daily on consecutive days? If so, specify the period.

L 117: Can you briefly report the procedure described by Yu and Morrison for the DNA extraction?

L 120: Can you briefly describe the procedure reported by Grimm et al.,?

L147: Can you briefly describe the procedure for the SCFAs extraction?

Results:

I believe that the results are accurate, the tables and figures are clear and adequately explained.

L 246: You have to correct “Fig 4” and put a period after the word “S4 Table”.

Discussion:

L 294-295: In addition to the environment in which the horse lives, another factor that influences the microbiome, in this case its fibrolytic activity, is the diet. I would suggest you to cite the study “Raspa, F., Vervuert, I., Capucchio, M. T., Colombino, E., Bergero, D., Forte, C., Greppi, M., Cavallarin, L., Giribaldi, M., Antoniazzi, S., Cavallini, D., Valvassori, E., & Valle, E. (2022). A high-starch vs. high-fibre diet: Effects on the gut environment of the different intestinal compartments of the horse digestive tract. BMC Veterinary Research, 18(1), 187. " ext-link-type="uri" xlink:type="simple">https://doi.org/10.1186/s12917-022-03289-2"

Furthermore, the diet has an effect on the gut histomorphometry with the possibility of repercussions on microbiome. In this regard I would suggest you cite another recent article “Colombino, E., Raspa, F., Perotti, M., Bergero, D., Vervuert, I., Valle, E., Capucchio, M. T. (2022). Gut health of horses: Effects of high fibre vs high starch diet on histological and morphometrical parameters. BMC Veterinary Research, 18(1), 338. https://doi.org/10.1186/s12917-022-03433-y ".

Since in your study the diet is the same for all the horses, this factor does not contribute to changes in the microbiome, as you pointed out for the environment.

L 353-356: Always in the previously mentioned study of “Raspa, F., Vervuert, I., Capucchio, M. T., Colombino, E., Bergero, D., Forte, C., Greppi, M., Cavallarin, L., Giribaldi, M., Antoniazzi, S., Cavallini, D., Valvassori, E., Valle, E. (2022). A high-starch vs. high-fibre diet: Effects on the gut environment of the different intestinal compartments of the horse digestive tract. BMC Veterinary Research, 18(1), 187. " ext-link-type="uri" xlink:type="simple">https://doi.org/10.1186/s12917-022-03289-2"

was reported how a diet rich in starch leads to differences in microbiomes’s fibrolytic activity. In particular, an increase in the production of total VFAs was observed in a high starch diet compared to a fibre rich diet. Moreover, an increase in the production of butyrate and valerate and a decrease in acetate was reported by feeding horses with large quantities of cereals in the diet.

L 312-314: With the phrase "ponies negative for Cushing's syndrome" do you mean healthy individuals? Rephrase this sentence.

L 371: “evolution” in “change”

Is there any conflict of interest?

Reviewer #2: Thank you for the opportunity to review this manuscript. The design is adequate and brings important novel information to this field. I have only minor suggestions on how the authors refer to their study. This is not a longitudinal study but a cross-sectional design.

Abstract: Is there any scientific evidence about the incidence of diseases in horses older than 15 years old, or just owners’ anecdotal perception? I recommend that you rephrase that to sound more science-based: “There is evidence…”

Re-define objective: it is hard to define health outcomes in only 50 horses. Furthermore, you did not find any health problems. I think your objective was to compare horses of different ages in the same environment.

The term “End-of-life” may not be appropriate, as many horses live up to 30 years. That would be more like middle age.

The study design is not clear in the abstract. Please state that samples were collected at one point in time.

It is inadequate to use the term “decreased” because you did not do longitudinal comparisons.

Conclusions: I believe any inference to pathologies should be in the discussion rather than in the conclusions.

Line 63: please don’t use the term “poorly”, as it can sound a bit detrimental to the work done by other researchers.

Line 68 and somewhere else: please don’t use the term ASV, as most of the readers are not familiar with the bioinformatic concepts. An unclassified member or a bacterial member of that family would be preferable.

Line 88: I would not classify your study as longitudinal.

Results: Please confirm that the horse populations (sex, breed) were evenly distributed across age groups.

Line 155: Please indicate which parameters you used to calculate the correlation.

Line 214: the species-level classification is confusing for readers. How can you find only 12 species? The use of short-read sequencing (Illumina) does not support reliable classification at the species level, even when sequencing the V3-V4 region. Please consider removing this information.

Line 246: Fig 4

Line 287: please rephrase “changes” by “differences.”

Line 395: I believe any inference to pathologies should be in the discussion rather than in the conclusions. Please consider modifying that in the abstract as well.

6. PLOS authors have the option to publish the peer review history of their article (what does this mean?). If published, this will include your full peer review and any attached files.

Reviewer #1: **Yes: **Emanuela Valle

Reviewer #2: **Yes: **Marcio Costa

---

## [Author Response · Author response to Decision Letter 0]

3 Apr 2024

Reviewer 1 : 

L 77 “evolution” in “change” - Taken into account.

L 77-78: I would change the sentence “while including parameters of microbial fibrolytic activity” in “and evaluated parameters of microbial fibrolytic activity”. - Taken into account.

L 88-97: Do horses, especially older ones, suffer from any metabolic disorder? Specify in the materials and methods that no tests have been carried out to investigate the possible presence of metabolic disorders in elderly subjects. - In order to be more precise we have added the following sentence in the manuscript “Potential metabolic disorders were not tested.”

L 99-100: Have you carried out analyzes of the hay and feed concentrate used? Is it possible to provide a table with the chemical composition of feeds and the nutritional values of the overall daily diet? - Hay and concentrate were analysed by DairyOne lab. We have added table in supplementary data.

L 105: Were faecal samples collected daily on consecutive days? If so, specify the period. - Faecal samples were collected in a single point. To avoid confusion, we removed "longitudinal study" and we had “After 3 weeks of constant diet, freshly voided faecal samples were obtained from each horse once on the same day.”

L 117: Can you briefly report the procedure described by Yu and Morrison for the DNA extraction? - We have added the following sentences to be more precise in the procedure: “Faecal total DNA was extracted as described by Yu and Morrison (43). Briefly, to lyse the cells, 0.25 g of faecal sample was bead-beaten with a mixture of sodium dodecyl sulfate (SDS), NaCl, and EDTA. Impurities and SDS were removed by precipitating them with ammonium acetate. Nucleic acids were then recovered by precipitating them with isopropanol. Genomic DNA was purified by sequentially digesting with RNase and Proteinase K, followed by the use of QIAamp columns.”

L 120: Can you briefly describe the procedure reported by Grimm et al.,?- We have added the following sentences to be more precise in the procedure: “Briefly, 2 consecutive polymerase chain reactions (PCR) were performed (PCR 1 for V3-V4 region amplification and PCR 2 to ligate Illumina adapters and index for sample identification). The PCR mix contained DNA, buffer, dNTPs, Taq polymerase and primers (PCR 1: F343 and R784; PCR 2: forward primer AATGATACGGCGACCACCGAGATCTACACTCTTTCCCTACACGAC and reverse primer CAAGCAGAAGACGGCATACGAGAT-Index-GTGACTGGAGT TCAGACGTGT). The PCR programme was as follows: 1 minute at 94°C, 30 (PCR 1) or 12 (PCR 2) x [94°C for 1 minute, 65°C for 1 minute and 72°C for 1 minute] and 10 minutes at 72°C. The correct V3-V4 region amplification after PCR 1 was verified by electrophoresis on a 2% agarose gel. The PCR products obtained were sequenced using an Illumina MiSeq run of 250-paired ends, according to the manufacturer’s instructions (Illumina Inc., San Diego, CA, United States) at Genotoul Bioinformatics Platform (Toulouse, France). FROGS (Find Rapidly OTU with Galaxy Solution) metabarcoding pipeline on the Galaxy server was used to performed bioinformatics analysis (45).”

L147: Can you briefly describe the procedure for the SCFAs extraction? - We have added the following sentences to be more precise in the procedure: “SCFAs concentrations were determined as described by Jouany (46). Briefly, filtered faecal samples were injected, under nitrogen, onto a 30 m x 0.25 mm diameter x 0.25 µm capillary column (Elite-FFAP column; PerkinElmer, Courtaboeuf, France) of gas-liquid chromatography coupled to a flame ionisation detector (Clarus; PerkinElmer, Courtaboeuf, France). The internal standard added to all filtered faecal samples was 4-methyl valeric acid (277827-25G, Sigma-aldrich, USA). A standard solution was used to determine the concentration of acetate (C2), propionate (C3), butyrate (C4), iso-butyrate (iC4), valerate (C5), and iso-valerate (iC5) in each filtered faecal sample. The addition of each SCFA gave the total SCFAs concentration. Each SCFA was expressed as a percentage of the total SCFAs.”

L 246: You have to correct “Fig 4” and put a period after the word “S4 Table”. - Taken into account.

L 294-295: In addition to the environment in which the horse lives, another factor that influences the microbiome, in this case its fibrolytic activity, is the diet. I would suggest you to cite the study https://doi.org/10.1186/s12917-022-03289-2. Furthermore, the diet has an effect on the gut histomorphometry with the possibility of repercussions on microbiome. In this regard I would suggest you cite another recent article https://doi.org/10.1186/s12917-022-03433-y. Since in your study the diet is the same for all the horses, this factor does not contribute to changes in the microbiome, as you pointed out for the environment. - We completely agree that diet is a major factor of variation for the digestive microbiome. This has been summarized in reviews. We amended the paragraph and added references number 51 and 52: “In humans, the country (50) and residence place (28) in which elderly people live, are described as a co-factor contributing to the observed differences in faecal bacterial communities. In horses, the location explains 6.4% of the variation in faecal bacterial communities (40), but not specifically in elderly individuals. Moreover, reviews summarise and highlight the major effect of the diet on the large intestine microbiota (51,52). Thus, to limit the impact of the location and diet and highlight only the effect of age on the faecal bacterial communities, we selected horses kept at one location for at least 1 year and fed the same diet.”

L 353-356: Always in the previously mentioned study https://doi.org/10.1186/s12917-022-03289-2 was reported how a diet rich in starch leads to differences in microbiomes’s fibrolytic activity. In particular, an increase in the production of total VFAs was observed in a high starch diet compared to a fibre rich diet. Moreover, an increase in the production of butyrate and valerate and a decrease in acetate was reported by feeding horses with large quantities of cereals in the diet. - We did not aim at measuring the effect of the diet as the horses were fed the same one along the experimental period. We measured the effect of age category. 

L 312-314: With the phrase "ponies negative for Cushing's syndrome" do you mean healthy individuals? Rephrase this sentence. - We cannot replace by healthy as we do not know if those ponies did not have other health issues.

L 371: “evolution” in “change” - Taken into account.

Reviewer 2: 

Is there any scientific evidence about the incidence of diseases in horses older than 15 years old, or just owners’ anecdotal perception? I recommend that you rephrase that to sound more science-based: “There is evidence…” - To our knowledge, no causal relationship has been established between age and disease prevalence. Owners' responses to questionnaires have been corroborated by veterinarians' clinical examinations in some studies (doi.org/10.1111/j.2042-3306.2010.00361.x;
doi.org/10.1016/j.tvjl.2011.03.021). We have added reference number 8 for the introduction part.

Re-define objective: it is hard to define health outcomes in only 50 horses. Furthermore, you did not find any health problems. I think your objective was to compare horses of different ages in the same environment. The term “End-of-life” may not be appropriate, as many horses live up to 30 years. That would be more like middle age. - We have redefined objective as recommended: “This study aimed to compare the large intestine microbiome in horses aged between 6 to 30 years old (yo), living in the same environment and consuming the same diet, in order to assess whether the parameters changed linearly with age and whether there was a pivotal age category.”. We have deleted consideration regarding health outcomes.

The study design is not clear in the abstract. Please state that samples were collected at one point in time. - We have given precision to be clearer: “After three weeks of constant diet (ad libitum hay and 860 g of concentrate per day), one faecal sample per horse was collected on the same day.”

It is inadequate to use the term “decreased” because you did not do longitudinal comparisons. - We agree that the word decrease is more appropriate when talking about a longitudinal study. We suggest that we remain factual and speak only of positive or negative correlation with age. We have applied this modification throughout the manuscript and removed all “ageing process” associated with it.

Conclusions: I believe any inference to pathologies should be in the discussion rather than in the conclusions. - We did not want to go into the pathology part in detail, we just wanted to give a possible application of the results we found. If we know which age group to target (at which major changes in the microbiome can occur), we can take preventive action to avoid reaching pathological states. We have reworded it to make it sound more like a perspective: “Our data suggest that the microbiome changes during the ageing process. By highlighting the pivotal age of 16-20, this gives the opportunity to intervene before individuals reach extremes that could lead to pathological conditions.”

Line 63: please don’t use the term “poorly”, as it can sound a bit detrimental to the work done by other researchers. - We have taken it into account. This was awkward translation. We meant there is little documentation.

Line 68 and somewhere else: please don’t use the term ASV, as most of the readers are not familiar with the bioinformatic concepts. An unclassified member or a bacterial member of that family would be preferable. - We cannot replace this term as it is the one used in the cited publication (Theelen et al., 2021). This result comes from a DESEQ2 analysis, which compares only ASVs. The graph in the cited publication only gives information up to the family level. To improve the reader's understanding, we have changed the term "belonging".

Line 88: I would not classify your study as longitudinal. - Taken into account.

Please confirm that the horse populations (sex, breed) were evenly distributed across age groups. - Below is a table summarising the distribution of sex and type of horse by age group (detailed in S1 table). There were geldings and females in all age groups. The distribution was not always 50/50. However, the effect of sex on faecal microbiota parameters has not always been demonstrated (doi.org/10.3390/ani11061762). In addition, when working with sanctuary horses, their breed was not always known. We therefore decided to classify them into types: Coldblood, Warmblood and Hotblood. There were no ponies. We wanted the horses to have been in the same environment for a long time (1 year) so we had to compromise on the distribution of sex and breed in the groups.

 Gelding (%) Female (%) Coldblood (%) Warmblood (%) Hotblood (%)

6 to 10 yo 70 30 100 0 0

11 to 15 yo 45 55 64 36 0

16 to 20 yo 85 15 0 54 46

21 to 30 yo 57 43 0 100 0

Line 155: Please indicate which parameters you used to calculate the correlation. - As we have correlated all parameters with age, we cannot list them all. We have rephrased and clarified our sentence: “Pearson correlations between all the parameters studied and age were obtained using the PROC CORR procedure of the SAS software.”

Line 214: the species-level classification is confusing for readers. How can you find only 12 species? The use of short-read sequencing (Illumina) does not support reliable classification at the species level, even when sequencing the V3-V4 region. Please consider removing this information. - We have removed the information.

Line 246: Fig 4 - Taken into account.

Line 287: please rephrase “changes” by “differences.” - We prefer keeping the term "changes" when talking about the linear study (correlation) and the term "differences" when talking about the study by category (ANOVA). For the linear study, we had also used the term "evolution" in our first draft. As the first reviewer asked us to modify it to "change" we now only use “change” along the text.

Line 395: I believe any inference to pathologies should be in the discussion rather than in the conclusions. Please consider modifying that in the abstract as well. - As explained for the abstract, we do not want to describe the pathological states in detail in the body of the manuscript. We only want to give an application of the results highlighted in this study. To make it sound more like a perspective, we have rephrased the conclusion.

---

## [Decision Letter · Decision Letter 1]

18 Apr 2024

Changes of faecal bacterial communities and microbial fibrolytic activity in horses aged from 6 to 30 years old

PONE-D-23-43107R1

Dear Dr. Baraille,

We’re pleased to inform you that your manuscript has been judged scientifically suitable for publication and will be formally accepted for publication once it meets all outstanding technical requirements.

Kind regards,

Franck Carbonero, PhD

Academic Editor

PLOS ONE

Additional Editor Comments (optional):

Reviewers' comments:

Reviewer's Responses to Questions

**Comments to the Author**

1. If the authors have adequately addressed your comments raised in a previous round of review and you feel that this manuscript is now acceptable for publication, you may indicate that here to bypass the “Comments to the Author” section, enter your conflict of interest statement in the “Confidential to Editor” section, and submit your "Accept" recommendation.

Reviewer #2: All comments have been addressed

2. Is the manuscript technically sound, and do the data support the conclusions?

Reviewer #2: Yes

3. Has the statistical analysis been performed appropriately and rigorously? 

Reviewer #2: Yes

4. Have the authors made all data underlying the findings in their manuscript fully available?

Reviewer #2: Yes

5. Is the manuscript presented in an intelligible fashion and written in standard English?

Reviewer #2: Yes

6. Review Comments to the Author

Reviewer #2: (No Response)

7. PLOS authors have the option to publish the peer review history of their article (what does this mean?). If published, this will include your full peer review and any attached files.

Reviewer #2: **Yes: **Marcio Costa

---

## [Editor Report · Acceptance letter]

24 May 2024

PONE-D-23-43107R1 

PLOS ONE

Dear Dr. Baraille, 

I'm pleased to inform you that your manuscript has been deemed suitable for publication in PLOS ONE. Congratulations! Your manuscript is now being handed over to our production team.

Kind regards, 

on behalf of

Dr. Franck Carbonero 

Academic Editor

PLOS ONE